# Nursing Student Knowledge Related to Sepsis in Croatian, Cypriot, and Greek Universities: A Cross-Sectional European Study

**DOI:** 10.3390/ijerph21070922

**Published:** 2024-07-15

**Authors:** Adriano Friganović, Gloria Bešker, Jelena Slijepčević, Kristian Civka, Sanja Ledinski Fićko, Sabina Krupa, Ana Brčina, Stelios Iordanou, Andreas Protopapas, Maria Hadjibalassi, Vasilios Raftopoulos, Theodoros Katsoulas

**Affiliations:** 1University Hospital Centre Zagreb, Kisptatićeva12, 10000 Zagreb, Croatia; gloriavalicevic@gmail.com (G.B.); jelena.slijepcevic.kbc@gmail.com (J.S.); kristian.civka@hdmsarist.hr (K.C.); ana.brcina28996@gmail.com (A.B.); 2Department of Nursing, University of Applied Health Sciences, Mlinarska Cesta 38, 10000 Zagreb, Croatia; sanja.ledinski-ficko@zvu.hr; 3Faculty of Health Studies, University of Rijeka, 51000 Rijeka, Croatia; 4Institute of Health Sciences, College of Medical Sciences, University of Rzeszow, 35-310 Rzeszow, Poland; sabciakr@gmail.com; 5Quality Assurance Department, Strate Health Services Organization, 4131 Limassol, Cyprus; iordanou.stelios@gmail.com; 6Department of Health Sciences, European University, 2404 Nicosia, Cyprus; a.protopapas@euc.ac.cy; 7Department of Nursing, School of Health Sciences, Cyprus University of Technology, 3036 Limassol, Cyprus; maria.hadjibalassi@cut.ac.cy; 8Hellenic National Public Health Organization, 15123 Athens, Greece; vraftop1@gmail.com; 9Department of Nursing, National and Kapodistrian University of Athens, 11527 Athens, Greece; tkatsoul@nurs.uoa.gr

**Keywords:** sepsis, education, students, nurses

## Abstract

Background: Although the treatment of sepsis has advanced during the past 20 years there is still a high incidence and high mortality, which make sepsis one of the leading public health problems. Adequate knowledge of sepsis and the sepsis guidelines is still the most important pillar for nurses because of the long time they spend with critically ill patients. Given their frontline role in patient care, nurses are pivotal in early sepsis recognition, timely intervention, and ensuring adherence to treatment protocols. Aim: This study aimed to investigate nursing students’ knowledge of sepsis and the symptoms of sepsis, and to compare the results of nursing students from several European universities (Croatia, Cyprus, Greece). Methods: A cross-sectional design was used, with a sample of 626 undergraduate nursing students from Croatian, Cypriot, and Greek universities from 2022 to 2023. Demographic features (gender, age, employment, year of study) and a questionnaire provided by Eitze et al. were utilized as instruments. Results: There was a statistically significant difference among the countries (F_(2.625)_ = 4.254, *p* = 0.015) in average knowledge about sepsis, with Scheffe’s post hoc test indicating that the Cypriot students had a higher average knowledge than the Greek students (*p* = 0.016), while students from neither country were significantly different from Croatian students (both *p* > 0.05). Conclusions: This study showed the still limited knowledge of nursing students and the differences among the educational programs for nursing students. The educational curricula of nursing studies should increase the number of sepsis lectures and use innovative techniques.

## 1. Introduction

Sepsis is an enormous global healthcare problem and a challenge for healthcare professionals everywhere [1]. It is one of the major causes of mortality and morbidity in almost all parts of the world [2]. Sepsis represents an organism’s response to infection, and if it is not managed properly, it can cause multiorgan failure and even death [3]. Although the treatment of sepsis has advanced during the past 20 years, there is still a high incidence and high mortality, making sepsis one of the leading public health problems [3]. Adequate knowledge of sepsis and the sepsis guidelines is still the most important pillar for nurses because of the long time they spend with critically ill patients [4].

In the healthcare system, nurses are the frontline responders in identifying and initiating care for patients with sepsis. Whether in the bustling emergency department or the attentive setting of the wards, nurses are often the first to detect signs of community-acquired or hospital-onset sepsis. Their vigilance and prompt action can make a significant difference in patient outcomes. Nurse-led initiatives focusing on sepsis screening have demonstrated tangible benefits, such as reducing mortality rates and improving overall sepsis care. Thus, it is paramount that nurses not only understand the critical nature of their role in sepsis recognition but also receive comprehensive training to identify sepsis symptoms swiftly and respond with confidence and competence [5].

Students’ knowledge is a very important indicator of their future work in healthcare settings when they complete their education [6]. A similar study was conducted in Croatia, which explored knowledge about sepsis among nursing students. According to the study’s research findings, it was revealed that undergraduate nursing students exhibited insufficient comprehension of sepsis, as indicated by the diverse and inaccurate responses regarding its etiology, pathogenesis, and symptomatology [6].

Therefore, it is crucial that nursing and medical graduates possess sufficient knowledge and competencies to identify and initiate appropriate management protocols for sepsis, given their primary role as initial caregivers in general healthcare settings. Consequently, it is imperative to ensure that undergraduate medical and nursing students receive comprehensive training to assess, identify, and manage sepsis cases effectively upon entering clinical practice. However, the existing literature indicates a deficiency in sepsis education among medical and nursing students, suggesting inadequate coverage and preparation in undergraduate curricula. Thus, there is a pressing need to develop educational programs tailored to equip future healthcare professionals with the necessary skills and knowledge to handle sepsis cases competently [7].

As member states of the European Union, Croatia, Greece, and Cyprus are obligated to ensure that their nursing education programs adhere to the minimum standards outlined in Directive 2005/36/EC. Despite variations in program duration, all three countries have harmonized their curricula to meet the essential requirements, thus ensuring uniformity and quality in nursing education across the EU. According to Directive 2005/36/EC, the minimum educational prerequisites for nursing programs include (a) a minimum of 10 years of general education, (b) a nursing training program comprising at least 3 years of study or 4600 h of combined theoretical and practical training, and (c) official recognition of these programs by the respective national authorities. Annex V.2, 5.2.1 of the directive further specifies that the training leading to the formal qualification of nurses responsible for general care, which all three countries have aligned with.

The Bachelor’s programs for nursing in Greece and Cyprus last for 4 years, while in Croatia, the program lasts for 3 years [6,8,9].

In Croatia, the nursing undergraduate curriculum integrates sepsis across various courses, covering its pathogenesis, therapeutic approaches, and nursing interventions. Specifically, students encounter sepsis topics in Microbiology during the first year, while Nursing Care of Adults I, Infectiology, and Internal Medicine courses address sepsis in the second year. Moreover, Nursing Care of Adults II delves further into sepsis management in the third year [6].

In Greece, nursing students encounter the concept of sepsis in their first year in subjects such as Anatomy and Histology, in their second year in courses such as Microbiology, Physiology I, Surgery–Physical Examination, Internal Medicine I, and Pathophysiology of Diseases, in their third year in subjects like Internal Medicine II, Surgery, Medical Nursing, Infection Control Nursing, and in their final year in subjects such as Pediatric Nursing, Emergency Medicine, and Intensive Care [8].

In Cyprus, nursing students encounter the concept of sepsis in their second year in subjects such as Pathophysiology, Microbiology, and Infection Control. In their third year, they further explore sepsis in microbiology courses, and in their fourth year, within the context of Intensive Care and Specialties. In their first year, they delve into courses like Anatomy–Physiology, Psychology, and Introduction to Nursing Science, as well as Anatomy Biology and Biochemistry, among others. The second year mainly focuses on Pathophysiology, Pharmacology, Nursing Informatics, and Medical and Surgical Nursing Specialties. By the third year, their focus shifts to Nursing Research, Nursing Management, Child, Midwifery, and Gynecological Nursing Pathophysiology, and in the fourth year, they engage in subjects such as Emergency Nursing, Intensive Care, and Epidemiology for Nursing [9].

The ability of nursing students to recognize and respond to a patient’s condition due to sepsis is very important, and education about sepsis is essential [6]. Bladon et al. found a very strong association between sepsis morbidity and mortality and the factors associated with health inequalities and inadequate education [10].

There is also the possibility for nursing schools to offer a teaching technique for students to learn about sepsis and the symptoms of sepsis. Some authors suggest using the Early Detection Sepsis Assessment Checklist, with the goal of improving nursing students’ recognition of sepsis through clinical simulation [11]. Martinez and Aronson concluded that simulation-based experience was highly effective in preparing students to care for patients with early signs of sepsis [12]. It is also important that nursing students are ready to ensure the health literacy of citizens and provide accurate information [13].

Gustad et al. stated that it is very important to improve the awareness of sepsis and enhance interprofessional collaboration among nurses, physicians, and other medical personnel [14]. However, medical staff usually rely on their clinical judgment as well as the use of objective measurements [13]. Continual repetition and education for new colleagues were identified as important factors for the sustainability of the intervention, and implementing a sepsis pathway accelerated the identification of sepsis and improved clinical outcomes [14,15,16,17,18]. A study conducted by Gustad et al. highlighted the importance of standardized protocols and training for the early detection and management of sepsis in healthcare settings [14].

Consequently, an educational approach that intertwines sepsis education with interprofessional team training can significantly bolster the knowledge and practice of sepsis care among healthcare professionals. Furthermore, the growing emphasis on interprofessional team training prior to licensure underscores the necessity of incorporating components such as interprofessional communication and teamwork into undergraduate interprofessional education programs [5]. Exploring nurses’ self-assurance in identifying and managing sepsis is crucial, given its impact on patient outcomes [19].

The present study aimed to investigate nursing students’ knowledge of sepsis and its symptoms and to compare the results of nursing students from three European countries (Croatia, Cyprus, Greece). The aim of our research was to compare the knowledge of students at different European universities. The next step of our further research will be a comparison of curriculum content with regard to the results of this study.

Even though earlier research was conducted in this field, it is still not completely clear whether the education curriculum for nursing students is adequate or if changes are necessary to improve the knowledge of nursing students. This study will bring new insights into the knowledge of sepsis and its symptoms by comparing the results of different universities.

## 2. Materials and Methods

### 2.1. Respondents and Procedure

A cross-sectional design was used. A sample of 626 undergraduate nursing students was taken from three European countries in at least one university in each country, Croatia (CRO), Cyprus (CY), and Greece (GR), during the 2022–2023 academic year. The sample was convenience-based, but participation was only from universities that responded to our research team. Of these universities, 416 participants were from Croatia, 127 from Cyprus, and 80 from Greece. In Croatia, the University of Applied Health Sciences participated in this study. In Cyprus, the following universities participated: Cyprus University of Technology, European University of Cyprus, and Frederic University. In Greece, data were collected from the National and Kapodistrian University of Athens. Regarding the sample size, an a priori calculation was performed. Considering that the main analysis used to obtain the answers to the set goals was the chi-square test for differences among categories, the required sample size was calculated with the following assumptions: 3 groups of subjects, 1 measurement point, moderate expected effect size, and a two-way significance level of *p* < 0.05. The GPower 3.1 program used to calculate the a priori expected power of the test (power) showed that N = 330 subjects was desirable for the expected power of 0.09. In the event that additional covariate variables were included in the analysis, it would be optimal to have N = 500. The post hoc power was >0.90.

All enrolled students were invited to participate. The authors wanted to reach as many students as possible, without limiting them to a specific year, knowledge level, etc. The inclusion criteria were that both part-time (those employed in the healthcare system) and full-time students were included. Participation was voluntary, and all students signed informed consent. At each university, the teaching staff explained the goals and the reasons for this research to the students before they completed the surveys.

### 2.2. Research Instruments

For this study, demographic features (gender, age, employment, year of study) were collected using a questionnaire provided by Eitze et al. [20]. With the author’s approval, two items in the questionnaire were improved [6,20]. The questionnaire was not previously validated. It was chosen because it is simple to use and understand, with clearly marked correct answers, which ensured wider usage in different countries.

The questionnaire (see Appendix A) consisted of two categories as follows: sociodemographic data, which included 6 questions, and questions related to sepsis knowledge, consisting of 18 questions to which respondents could answer “yes”, “no”, or “not sure”.

The questionnaire was translated into Greek by a group of experts and then back-translated into English by a bilingual nurse to ensure alignment with the original questionnaire. This process was deemed satisfactory, with minimal differences observed. Similarly, in Croatia, a double translation was conducted from English to Croatian, and vice versa, revealing no significant discrepancies. Besides linguistics, psychometric validation was performed for all three languages. The questionnaire was distributed online in all three countries. A panel of experts in all three countries face-validated the contents prior to the first usage, and it was estimated to give good face validity.

The post hoc validation showed relatively good internal consistency for both subscales. In Cyprus, the Cronbach alpha for the knowledge scale was 0.64, and for the symptoms scale, it was 0.76; in Greece, the Cronbach alpha for the knowledge scale was 0.60, and for the symptoms scale, it was 0.73; and in Croatia, the Cronbach alpha for the knowledge scale was 0.53, and for the symptoms scale, it was 0.60.

These are satisfactory levels of reliability, except in Croatia, where the item are a skin rash and eczema symptoms of sepsis? was problematic. Without this item, reliability would increase to 0.73.

In order to verify the factor structure, a factor analysis of the main components with varimax rotation was performed. It was performed for the Croatian subsample, which had the largest sample size. The correlation matrix proved suitable for factor extraction (Kaiser–Meyer–Olkin measure, KMO = 0.713). The number of factors was determined based on the Guttman–Kaiser criterion (the number of factors according to the criterion that their eigenvalue, i.e., the value of the characteristic root exceeds 1, which is also visible in the scree plot diagram) and the percentage of explained variance of each subsequent factor. The first factor explained most of the variance in source of stress, at 16%. The next factor explained 7% (their characteristic roots were 3.01 and 1.39).

### 2.3. Data Analysis

The data analysis was carried out using SPSS (IBM, V 25.0) [21]. For the nominal (categorical) variables, the number and percentage of participants are shown, while the statistical significance among the variables was calculated using the chi-square test, with Fisher’s exact p for 2 × 2 tables. Mean +/− standard deviation are shown for the normally distributed continuous variables, and the median and interquartile range are shown for variables deviating from a normal distribution. Data normality was tested by the Shapiro–Wilk test. Levene’s test was used to test the homogeneity of variance as a prerequisite for a one-way ANOVA, which tested differences in the average knowledge and symptoms scales among the countries.

### 2.4. Ethical Considerations

The researchers who performed this study followed all the recommendations and principles of the Declaration of Helsinki [22]. The study protocol was approved by the Ethical Committee of the University of Applied Health Sciences on 12 September Class: 602-03/22-18/540, Reg. No. 251-379-10-22-02. The bioethics protocol approval number from the University of Athens in Greece is 406/2022, dated 3 June. The national bioethics committee approval number from Cyprus is EΕΒΚ ΕΠ 2022.01.46. The students participated voluntarily and were required to sign informed consent; they were informed that they could withdraw from this study at any stage.

## 3. Results

The respondents were 626 undergraduate nursing students, of which 547 (87.4%) were female and 79 (12.6%) were male. Students from all three years of study in Croatia and four years in Cyprus and Greece were included. The sample is described in more detail in the table below (Table 1). The chi-square test showed that there were more Croatian students in the category 24–30 years, and in both Greece and Croatia, there were more older students (31+) than in Cyprus (χ^2^ = 107.014, df = 10, *p* < 0.001).

Table 2 shows the number and percentage of respondents with correct and incorrect or unsure answers to questions about sepsis.

There were several statistically significant differences in the students’ knowledge of sepsis in the three countries. In response to the item “With sepsis, you must call the emergency services immediately”, more students in Croatia gave the correct response than in the other two countries (χ^2^ = 6.867, df = 2, *p* = 0.032). For “Sepsis is an intense allergic reaction”, the students in Greece and Croatia had a higher percentage of correct answers (χ^2^ = 6.520, df = 2, *p* = 0.038), but for “Sepsis can be diagnosed by a red line infiltrating from a wound up to the heart”, the Greek and Cypriot students had more correct answers than the Croatians (χ^2^ = 12.143, df = 2, *p* = 0.002). In response to the item “There are more cases of breast cancer than cases of sepsis’” Cyprus had the highest percentage of students who answered correctly, while Greece had the lowest (χ^2^ = 23.435, df = 2, *p* < 0.001). For “Sepsis can be caused by influenza”, again, Cyprus had the highest percentage of students answering correctly, and Greece had the lowest (χ^2^ = 7.220, df = 2, *p* = 0.027).

If the correct answers to the questions about knowledge of sepsis were computed as a total score for each student, then it was possible to calculate the average score of correct answers for each country. A one-way ANOVA was used to calculate the significance of the difference in average results across the countries, as Levene’s test of homogeneity of variance was not statistically significant (*p* = 0.064) (Table 2).

There was a statistically significant difference among the countries (F_(2.625)_ = 4.254, *p* = 0.015) in the average score in knowledge about sepsis, with Scheffe’s post hoc test indicating that students in Cyprus had a higher average knowledge than those in Greece (*p* = 0.016), although these two countries are not significantly different from Croatia (both *p* > 0.05) (Figure 1).

Figure 1 shows the mean scores for items about the knowledge of sepsis with appropriate error bars for each country. One-way ANOVA was used to test the differences among countries since Levene’s tests showed adequate homogeneity of 0.064 for the knowledge subscale. ANOVA showed a statistically significant difference between countries in knowledge (F_(2.625)_ = 4.254, *p* < 0.015), and the post hoc test revealed that only Cyprus students had a higher average score than Greece.

Table 3 shows the number and percentage of respondents with correct and incorrect or unsure answers to questions about the symptoms of sepsis, along with *p* for the chi-square analysis (Table 3). There were several statistically significant differences among the students in the three countries with regard to the symptoms of sepsis. More students in Croatia gave the correct response to disorientation compared with students in the other two countries (χ^2^ = 18.547, df = 2, *p* < 0.001); more students in Cyprus gave the correct response to shortness of breath compared with students in Croatia (χ^2^ = 19.944, df = 2, *p* < 0.001); and more students in Greece and Cyprus gave the correct response to skin rash and eczema compared with students in Croatia (χ^2^ = 6.197, df = 2, *p* = 0.045).

If the correct answers to the questions about the symptoms of sepsis were computed as a total score for each student, then it was possible to calculate the average score of correct answers for each country (score “symptoms”). A one-way ANOVA was used to calculate the significance of the difference in average results across the countries, as Levene’s test of homogeneity of variance was not statistically significant (*p* = 0.318) (Table 3).

There was no statistically significant difference among the countries (F_(2.625)_ = 1.591, *p* = 0.205) in the average score for the symptoms of sepsis (Figure 2).

Figure 2 shows the mean scores for the items on the knowledge about the symptoms of sepsis with appropriate error bars for each country. One-way ANOVA was used to test for the differences among the countries since Levene’s tests showed adequate homogeneity 0.318 for the symptoms subscale.

Regarding the interlevel differences in Cyprus, it was found that there is a statistically significant difference in knowledge among years of study (F_(3.126)_ = 9.156, *p* < 0.001), with Scheffe post hoc test showing that students who studied for four years (*p* < 0.001) have better knowledge than first-grade students (Figure 3).

In Greece, one student from the first year was excluded. It was found that there is a statistically significant difference in knowledge among years of study (F_(2.78)_ = 4.475, *p* = 0.015), with the Scheffe post hoc test showing that students who studied three years have better knowledge than second-grade students (*p* = 0.023) (Figure 3).

In Croatia, only three years were included, and it was found that there is a statistically significant difference in knowledge among years of study (F_(2.416)_ = 7.385, *p* < 0.001), with the Scheffe post hoc test showing that students of the second (*p* = 0.004) and third years (*p* = 0.004) have better knowledge than first-grade students (Figure 3).

Figure 3 shows the mean scores for the items on the knowledge about sepsis with appropriate error bars for each year and each country. Since there were no data for the first year in Greece or for the fourth year in Croatia, separate one-way ANOVAs were used to test the differences among years of study. In Cyprus, average knowledge is highest among four-year students (F_(3.126)_ = 9.156, *p* < 0.001), in Greece, higher scores are achieved by third- and fourth-year students compared with second-year students (F_(2.78)_ = 4.475, *p* < 0.001), and in Croatia, second- and third-year students have higher average score for knowledge compared with first-year students (F_(2.418)_ = 7.385, *p* < 0.001).

## 4. Discussion

This study aimed to investigate nursing students; knowledge of sepsis and the symptoms of sepsis and to compare the results of nursing students from three European countries (CRO, CY, GR). The results showed that students in Cyprus had a statistically significantly higher level of knowledge than students in Greece *p* = 0.015 (Table 2), while there were no significant differences in comparing Cyprus and Greece to Croatia (*p* < 0.05). These findings indicate that it would be interesting for future research to compare different study programs. The sample results indicated that the students in Croatia and Greece were older than those in Cyprus (*p* < 0.001), which may explain the better results of the Cyprus sample according to Greece, where younger students were more capable of learning (Table 1).

Parson Leight et al. suggested that educational training and initiatives should focus on infection prevention and should include a high amount of media strategies [19] to achieve better results in students’ knowledge of sepsis and its symptoms (Table 2 and Figure 1) [23]. Our results showed differences in nursing students’ knowledge, which reveals the need for future research and improvements in educational curricula (Figure 1). Further work could explore whether efforts to raise students’ awareness of sepsis can reduce the incidence of sepsis and improve the early recognition of its symptoms and signs [23].

In Australia, Peters et al. found considerable knowledge gaps in parental awareness and knowledge of sepsis, particularly sepsis recognition, indicating that problems with health literacy could be improved by better nursing student knowledge and better involvement in raising awareness [13,24]. Nurses should facilitate parental education and target these knowledge gaps to improve health behavior [24]. Seymore et al. investigated knowledge of emergency services and also found an insufficient knowledge level of sepsis and the symptoms of sepsis (Table 2 and Table 3) [25]. They concluded that if paramedics could be integrated into strategies for the early identification and treatment of sepsis, the healthcare system would benefit from their education and training [25]. In a scoping review, Fiest et al. explored healthcare professionals’ awareness and knowledge of sepsis [26]. Their results suggested that patient/public awareness of sepsis has gradually improved over time and that awareness and knowledge vary globally [26,27]. In an interprofessional study, Metelman et al. found that emergency personnel rated the early initiation of sepsis treatment as important, but sepsis knowledge was limited, similar to the results of our study (Figure 2) [28]. Even though the majority of medical personnel from emergency departments had attended educational programs on sepsis within the past year, a high percentage of paramedics and emergency dispatchers had never received any training on sepsis [28]. In another study, Baez et al. confirmed a poor understanding of the principles of diagnosis and management of sepsis and suggested a need for enhanced education [29,30]. Salameh et al. claimed that medical personnel need sepsis management information provided by continual education programs and that the development of an approved protocol can improve nurses’ knowledge, attitudes, and practices [31].

This study’s limitations include the potential restriction imposed by the sample size, as the number of participants from each country might not comprehensively represent the entire population of undergraduate nursing students. Additionally, the findings’ generalizability may be constrained, given that this study was conducted exclusively in three specific European countries, raising questions about their applicability to broader regions or populations. Moreover, the unequal sample sizes among countries could introduce biases in the comparative analysis, potentially affecting the robustness of the conclusions drawn. To address these limitations in future research, efforts should be made to ensure more balanced sample sizes across participating countries and to include a more diverse range of regions or populations to enhance the generalizability of the findings. In all countries, the sample was mixed across age (year of study) and gender. Unfortunately, factors like exposure to sepsis education or clinical experience with sepsis patients were not controlled, so this remains the weak point, causing potential bias.

Additionally, employing longitudinal study designs and implementing specific educational interventions could provide further insights into the dynamics of sepsis knowledge among nursing students over time and the effectiveness of targeted educational approaches.

## 5. Conclusions

This study showed that nursing students in this sample still have limited knowledge of sepsis and that there are differences among education programs for nursing students.

These findings suggest that variations in educational programs or teaching methodologies at the universities included in this research may contribute to the differences in students’ knowledge levels. Additionally, the age distribution of students within the samples may play a role, as indicated by the observation that students in Croatia and Greece were older than those in Cyprus.

Education curricula of nursing studies should increase the number of lectures on sepsis to increase knowledge. For future research, it would be beneficial to conduct a longitudinal study with specific educational interventions. This highlights the importance of further research to explore the underlying factors influencing the differences in knowledge levels among nursing students from different universities.

## Figures and Tables

**Figure 1 ijerph-21-00922-f001:**
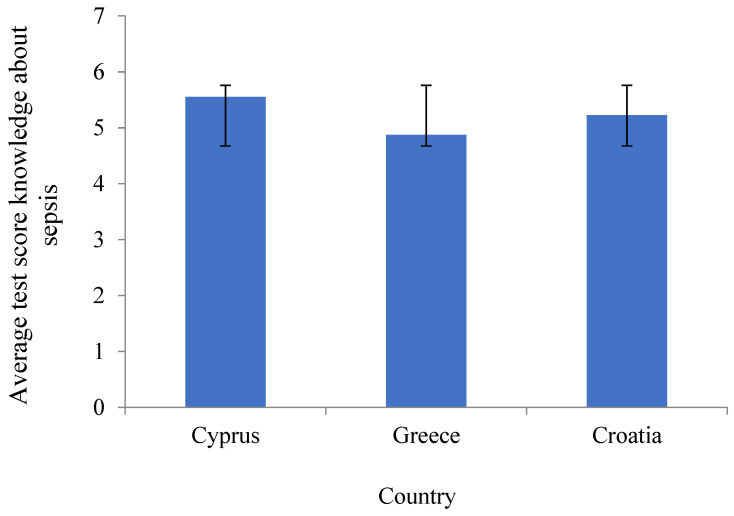
Average levels of “knowledge of sepsis” in the three countries.

**Figure 2 ijerph-21-00922-f002:**
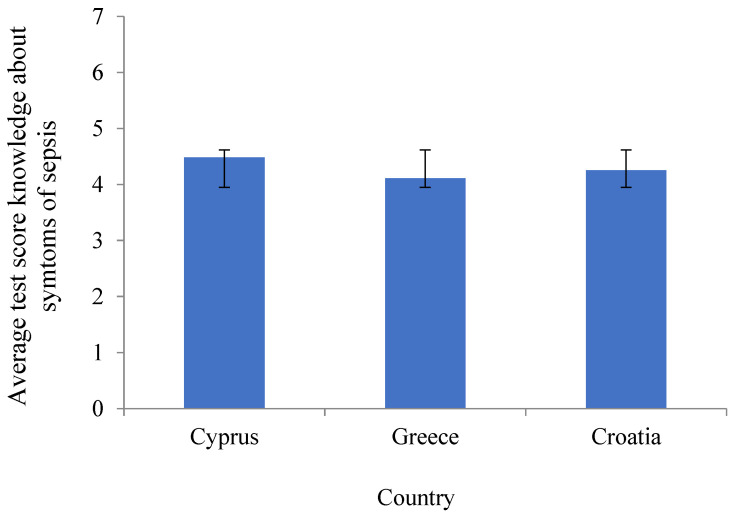
Average levels of “knowledge about the symptoms of sepsis” in the three countries.

**Figure 3 ijerph-21-00922-f003:**
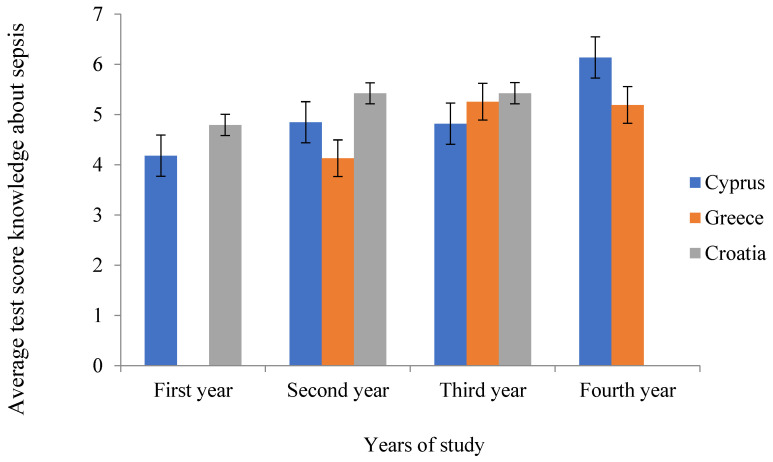
Differences among year levels of nursing students in their knowledge of sepsis.

**Table 1 ijerph-21-00922-t001:** Respondents’ demographic descriptions according to country.

		Country						Total		
		Cyprus		Greece		Croatia		n	(%)	
		n	(%)	n	(%)	n	(%)			*p*
Gender	Male	20	(15.7%)	9	(11.3%)	50	(11.9%)	79	(12.6%)	0.486
	Female	107	(84.3%)	71	(88.8%)	369	(88.1%)	547	(87.4%)	
Total		127	(100.0%)	80	(100.0%)	419	(100.0%)	626	(100.0%)	
Age	18–19	22	(17.3%)	4	(5.0%)	41	(9.8%)	67	(10.7%)	<0.001
	20	11	(8.7%)	27	(33.8%)	77	(18.4%)	115	(18.4%)	
	21	38	(29.9%)	9	(11.3%)	57	(13.6%)	104	(16.6%)	
	22–23	49	(38.6%)	22	(27.5%)	75	(17.9%)	146	(23.3%)	
	24–30	6	(4.7%)	5	(6.3%)	104	(24.8%)	115	(18.4%)	
	31 and older	1	(.8%)	13	(16.3%)	65	(15.5%)	79	(12.6%)	
Total		127	(100.0%)	80	(100.0%)	419	(100.0%)	626	(100.0%)	
Year of study	First year	22	(17.3%)	1	(1.3%)	131	(31.3%)	154	(24.6%)	--
	Second year	13	(10.2%)	23	(28.7%)	147	(35.1%)	183	(29.2%)	
	Third year	11	(8.7%)	35	(43.8%)	141	(33.7%)	187	(29.9%)	
	Fourth year	81	(63.8%)	21	(26.3%)			102	(16.3%)	
Total		127	(100.0%)	80	(100.0%)	419	(100.0%)	626	(100.0%)	

*p* < 0.05.

**Table 2 ijerph-21-00922-t002:** Knowledge of sepsis according to country.

		Country						Total		
		Cyprus		Greece		Croatia		n	(%)	
		n	(%)	n	(%)	n	(%)			*p*
With sepsis, you must call the emergency services immediately.	Incorrect or not sure	26	(20.5%)	20	(25.0%)	60	(14.3%)	106	(16.9%)	0.032
	Correct	101	(79.5%)	60	(75.0%)	359	(85.7%)	520	(83.1%)	
Total		127	(100.0%)	80	(100.0%)	419	(100.0%)	626	(100.0%)	
Sepsis is an intense allergic reaction.	Incorrect or not sure	31	(24.4%)	10	(12.5%)	66	(15.8%)	107	(17.1%)	0.038
	Correct	96	(75.6%)	70	(87.5%)	353	(84.2%)	519	(82.9%)	
Total		127	(100.0%)	80	(100.0%)	419	(100.0%)	626	(100.0%)	
Sepsis is an intense immune response of the body.	Incorrect or not sure	29	(22.8%)	19	(23.8%)	106	(25.3%)	154	(24.6%)	0.837
	Correct	98	(77.2%)	61	(76.3%)	313	(74.7%)	472	(75.4%)	
Total		127	(100.0%)	80	(100.0%)	419	(100.0%)	626	(100.0%)	
Sepsis is caused by multidrug-resistant superbugs in hospitals.	Incorrect or not sure	11	(8.7%)	4	(5.0%)	40	(9.5%)	55	(8.8%)	0.420
	Correct	116	(91.3%)	76	(95.0%)	379	(90.5%)	571	(91.2%)	
Total		127	(100.0%)	80	(100.0%)	419	(100.0%)	626	(100.0%)	
Sepsis can be diagnosed by a red line infiltrating from a wound up to the heart.	Incorrect or not sure	90	(70.9%)	57	(71.3%)	348	(83.1%)	495	(79.1%)	0.002
	Correct	37	(29.1%)	23	(28.7%)	71	(16.9%)	131	(20.9%)	
Total		127	(100.0%)	80	(100.0%)	419	(100.0%)	626	(100.0%)	
Mortality after heart attacks is higher than mortality after sepsis.	Incorrect or not sure	89	(70.1%)	66	(82.5%)	310	(74.0%)	465	(74.3%)	0.134
	Correct	38	(29.9%)	14	(17.5%)	109	(26.0%)	161	(25.7%)	
Total		127	(100.0%)	80	(100.0%)	419	(100.0%)	626	(100.0%)	
There are more cases of breast cancer than cases of sepsis.	Incorrect or not sure	72	(56.7%)	71	(88.8%)	276	(65.9%)	419	(66.9%)	<0.001
	Correct	55	(43.3%)	9	(11.3%)	143	(34.1%)	207	(33.1%)	
Total		127	(100.0%)	80	(100.0%)	419	(100.0%)	626	(100.0%)	
Sepsis can be caused by lung inflammation.	Incorrect or not sure	29	(22.8%)	30	(37.5%)	131	(31.3%)	190	(30.4%)	0.064
	Correct	98	(77.2%)	50	(62.5%)	288	(68.7%)	436	(69.6%)	
Total		127	(100.0%)	80	(100.0%)	419	(100.0%)	626	(100.0%)	
Sepsis can be caused by influenza.	Incorrect or not sure	61	(48.0%)	53	(66.3%)	244	(58.2%)	358	(57.2%)	0.027
	Correct	66	(52.0%)	27	(33.8%)	175	(41.8%)	268	(42.8%)	
Total		127	(100.0%)	80	(100.0%)	419	(100.0%)	626	(100.0%)	

*p* < 0.05.

**Table 3 ijerph-21-00922-t003:** Knowledge of the symptoms of sepsis according to country.

		Country						Total		
		Cyprus		Greece		Croatia		n	(%)	
		n	(%)	n	(%)	n	(%)			*p*
Are chills and fever symptoms of sepsis?	Incorrect or not sure	18	(14.2%)	11	(13.8%)	49	(11.7%)	78	(12.5%)	0.709
	Correct	109	(85.8%)	69	(86.3%)	370	(88.3%)	548	(87.5%)	
Total		127	(100.0%)	80	(100.0%)	419	(100.0%)	626	(100.0%)	
Is disorientation a symptom of sepsis?	Incorrect or not sure	52	(40.9%)	37	(46.3%)	110	(26.3%)	199	(31.8%)	<0.001
	Correct	75	(59.1%)	43	(53.8%)	309	(73.7%)	427	(68.2%)	
Total		127	(100.0%)	80	(100.0%)	419	(100.0%)	626	(100.0%)	
Is shortness of breath a symptom of sepsis?	Incorrect or not sure	37	(29.1%)	32	(40.0%)	214	(51.1%)	283	(45.2%)	<0.001
	Correct	90	(70.9%)	48	(60.0%)	205	(48.9%)	343	(54.8%)	
Total		127	(100.0%)	80	(100.0%)	419	(100.0%)	626	(100.0%)	
Is a high heart rate a symptom of sepsis?	Incorrect or not sure	27	(21.3%)	22	(27.5%)	73	(17.4%)	122	(19.5%)	0.097
	Correct	100	(78.7%)	58	(72.5%)	346	(82.6%)	504	(80.5%)	
Total		127	(100.0%)	80	(100.0%)	419	(100.0%)	626	(100.0%)	
Is low blood pressure a symptom of sepsis?	Incorrect or not sure	45	(35.4%)	34	(42.5%)	171	(40.8%)	250	(39.9%)	0.490
	Correct	82	(64.6%)	46	(57.5%)	248	(59.2%)	376	(60.1%)	
Total		127	(100.0%)	80	(100.0%)	419	(100.0%)	626	(100.0%)	
Is diarrhea a symptom of sepsis?	Incorrect or not sure	75	(59.1%)	55	(68.8%)	277	(66.1%)	407	(65.0%)	0.260
	Correct	52	(40.9%)	25	(31.3%)	142	(33.9%)	219	(35.0%)	
Total		127	(100.0%)	80	(100.0%)	419	(100.0%)	626	(100.0%)	
Are a skin rash and eczema symptoms of sepsis?	Incorrect or not sure	66	(52.0%)	40	(50.0%)	258	(61.6%)	364	(58.1%)	0.045
	Correct	61	(48.0%)	40	(50.0%)	161	(38.4%)	262	(41.9%)	
Total		127	(100.0%)	80	(100.0%)	419	(100.0%)	626	(100.0%)	

*p* < 0.05.

## Data Availability

The data in this study are available from the corresponding authors upon request. The data are not available to the public for confidentiality reasons.

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
