# Peer review of "Nursing Student Knowledge Related to Sepsis in Croatian, Cypriot, and Greek Universities: A Cross-Sectional European Study"

_ijerph, 2024, doi:10.3390/ijerph21070922_

Round 1
Reviewer 1 Report (Previous Reviewer 1)
Comments and Suggestions for Authors
In detail, I read and analyzed the manuscript Nursing Student Knowledge Related to Sepsis in Three European Countries: A Cross-Sectional Study.
The topic is up-to-date because of the well-known impact of sepsis on patient safety and hospital costs in every healthcare system.
I have already reviewed this manuscript, and after the authors' corrections, it is now clear, comprehensive, relevant to the field, and contains an IMRAD structure.
Introduction
The authors presented the importance of nursing students' knowledge about sepsis and gave a brief overview of nursing educational programs in the countries participating in the research. The introduction follows a funnel approach and ends by stating the study's objectives.
Materials and Methods
The methodology adequately describes the questionnaire collection method, the sampling technique in all three countries, and the sample size determination.
The research instrument is well presented, as is the data analysis. Ethical consent from all three countries is provided.
Results
Results are presented textually in five tables and interpreted appropriately and consistently throughout the manuscript. The table and legend are appropriate and easy to analyze and understand.
Discussion
The authors explained all the obtained results in the discussion. Relevant and actual sources of information support all listed facts. Potential limitations but strengths of the study are outlined in this section of the manuscript.
Conclusions
The conclusions align with the study's aims and are stated correctly.
Reference
The references used are up-to-date and properly cited.
Author Response
Thank you for your comprehensive analysis, and your comments for the manuscript.
Your suggestions made this manuscript better.
Reviewer 2 Report (Previous Reviewer 2)
Comments and Suggestions for Authors
The study presented by Friganović and colleagues examines an important aspect of Nursing education regarding a critical topic, which is sepsis.
Upon critical evaluation of the manuscript, one could clearly appreciate the following:
1- The multi-national aspect of the study, which covers range of European countries, rather than a specific neoughbouring countries.
2- The heterogeniety of the study population, in which students from across year levels were sampled.
However, major concerns were raised during the review of this manuscript which should be addressed before it can be recommended for publication.
a. All the results in this study was presented in table format, which is counterintuitive especially for the type of data presented. For example, Tables 3 and 5 present numerical values, which would be better presented in a graph (bar charts or dot plots).
b. Moreover, the authors indicated that specific descriptive statistics were used to describe the data based on its distribution. However, in tables 3 and 5 authors seems to present everything indiscriminately. Authors should adhere to the methods stated in their data analysis section and only present relevant data.
c. Relating to point No.2 above, one would expect to see differences, between year levels of Nursing students, in their knowledge of sepsis. However, this seems to be missing from the analysis and results section. Authors should clearly present such a finding, preferrably in a figure format.
I strongly believe that addressing these comments would significantly improve the manuscript and its presentation to the wide readership of the journal.
Author Response
Comment 1: The study presented by Friganović and colleagues examines an important aspect of Nursing education regarding a critical topic, which is sepsis.
Upon critical evaluation of the manuscript, one could clearly appreciate the following:
1- The multi-national aspect of the study, which covers range of European countries, rather than a specific neoughbouring countries.
2- The heterogeniety of the study population, in which students from across year levels were sampled.
Answer 1: Thank you for your comments, highly appreciated.
Comment 2: All the results in this study was presented in table format, which is counterintuitive especially for the type of data presented. For example, Tables 3 and 5 present numerical values, which would be better presented in a graph (bar charts or dot plots).
Answer 2: We replaced tables with figures, as suggested, presented in attached document.
Comment 3: Moreover, the authors indicated that specific descriptive statistics were used to describe the data based on its distribution. However, in tables 3 and 5 authors seems to present everything indiscriminately. Authors should adhere to the methods stated in their data analysis section and only present relevant data. Relating to point No.2 above, one would expect to see differences, between year levels of Nursing students, in their knowledge of sepsis. However, this seems to be missing from the analysis and results section. Authors should clearly present such a finding, preferrably in a figure format.
Answer 3: Tables 3 and 5 have been presented as figures, so tables will no longer be shown.
Comment 4: I strongly believe that addressing these comments would significantly improve the manuscript and its presentation to the wide readership of the journal.
Answer 4: Thank you for your valuable comments.

Reviewer 3 Report (New Reviewer)
Comments and Suggestions for Authors
Improvements:
The questionnaire used for assessing sepsis knowledge was not previously validated, raising concerns about its reliability and validity. This affects the credibility of the data collected and the overall findings of the study.
Recommendation:
Post-Hoc Validation: While the study is already completed, a post-hoc validation of the questionnaire can still be conducted to strengthen the reliability and validity of the findings.
Content Validity: Engage a panel of sepsis and nursing education experts to review the questionnaire items. They should evaluate whether the items comprehensively cover the relevant aspects of sepsis knowledge. Document their feedback and any subsequent revisions to the questionnaire.
Construct Validity: Perform exploratory factor analysis (EFA) on the existing data to identify underlying constructs measured by the questionnaire. If possible, conduct confirmatory factor analysis (CFA) on a separate sample to confirm the factor structure.
Reliability: Assess internal consistency using Cronbach's alpha to measure how well the items in the questionnaire are correlated. A value above 0.7 is generally acceptable. Conduct a test-retest reliability assessment if possible by re-administering the questionnaire to a subset of participants and calculating the correlation between the two sets of responses.
Report Validation: Document and include the results of the post-hoc validation study in the manuscript. This will demonstrate that the questionnaire is a reliable and valid tool for assessing sepsis knowledge.
There are potential inconsistencies in the reporting of statistical methods and results. The power analysis description is unclear and may not accurately reflect the study design.
Recommendation:
Statistical Clarity: Ensure that all statistical methods used are clearly described in the methods section. Provide detailed explanations for the choice of tests and the assumptions underlying them.
Consistency in Reporting: Check for and resolve any inconsistencies in the presentation of results. All statistical outcomes should be reported with appropriate measures of significance (e.g., p-values, confidence intervals).
Power Analysis: Clarify the power analysis conducted prior to the study. Ensure it is accurately described and reflects the expected effect sizes, sample sizes, and significance levels. This helps justify the sample size and strengthens the study's methodological rigor.
Further Controls:
Ensure the demographic characteristics of the sample are well-balanced across different countries to avoid potential confounding variables.
Include controls for potential biases, such as prior exposure to sepsis education or clinical experience with sepsis patients.
Consistency Issues:
The conclusions drawn about the differences in sepsis knowledge between countries are not fully supported by a rigorous analysis due to the limitations of the sampling method and questionnaire validation.
The interpretation of statistical results should be more cautious, given the potential biases introduced by the convenience sampling.
Main Questions Addressed:
The study adequately addresses the main question of comparing sepsis knowledge across countries. However, the methodology's limitations weaken the strength of the conclusions.
Specific experiments or analyses (e.g., factor analysis for questionnaire validation, more detailed demographic analysis) should be included to strengthen the conclusions.
Comments on the Quality of English Language
The English used in the manuscript is appropriate but could benefit from further refinement.
Author Response
Comment 1:
Improvements:
The questionnaire used for assessing sepsis knowledge was not previously validated, raising concerns about its reliability and validity. This affects the credibility of the data collected and the overall findings of the study.
Recommendation:
Post-Hoc Validation: While the study is already completed, a post-hoc validation of the questionnaire can still be conducted to strengthen the reliability and validity of the findings.
Content Validity: Engage a panel of sepsis and nursing education experts to review the questionnaire items. They should evaluate whether the items comprehensively cover the relevant aspects of sepsis knowledge. Document their feedback and any subsequent revisions to the questionnaire.
Answer 1:
Post hoc validation shows relatively good internal consistency for both subscales: in Cyprus, Cronbach alpha for Knowledge scale is 0.64, for Symptoms scale 0.76; in Greece Cronbach alpha for Knowledge scale is 0.60, for Symptoms scale 0.73; in Croatia Cronbach alpha for Knowledge scale is 0.53, for Symptoms scale 0.60.
These are satisfactory levels of reliability, except in Croatia where item Are a skin rash and eczema symptoms of sepsis? Is problematic, and without it reliability would increase to 0.73.
Panel of experts in all three countries face-validated the contents prior to first usage and was estimated to give good face validity.
Comment 2:
Construct Validity: Perform exploratory factor analysis (EFA) on the existing data to identify underlying constructs measured by the questionnaire. If possible, conduct confirmatory factor analysis (CFA) on a separate sample to confirm the factor structure.
Reliability: Assess internal consistency using Cronbach's alpha to measure how well the items in the questionnaire are correlated. A value above 0.7 is generally acceptable. Conduct a test-retest reliability assessment if possible, by re-administering the questionnaire to a subset of participants and calculating the correlation between the two sets of responses.
Report Validation: Document and include the results of the post-hoc validation study in the manuscript. This will demonstrate that the questionnaire is a reliable and valid tool for assessing sepsis knowledge.
Answer 2: In order to verify the factor structure of the, a factor analysis of the main components with varimax rotation was performed. It was done for Croatian subsample which has largest sample size. The correlation matrix proved suitable for factor extraction (Kaiser-Meyer-Olkin measure, KMO = 0.713). The number of factors was determined based on the Guttman-Kaiser criterion (the number of factors according to the criterion that their eigenvalue, i.e., the value of the characteristic root exceeds 1, which is also visible in the scree plot diagram) and the percentage of explained variance of each subsequent factor. The first factor explains most of the variance of Source of Stress, 16%. The next factors explains 7% (their characteristic roots: 3.01, 1.39). Tables presented in document attached. We didn't add tables to main manuscript if you find necessary we will change it.
Comment 3: Ensure the demographic characteristics of the sample are well-balanced across different countries to avoid potential confounding variables. Include controls for potential biases, such as prior exposure to sepsis education or clinical experience with sepsis patients.
Answer 3: In all countries, sample was mixed across age (year of study) and gender. Unfortunately, factors like exposure to sepsis education or clinical experience with sepsis patients were not controlled, so this remains the week point, causing potential bias and we will add this to study limitations.
Comment 4:
Consistency Issues: The conclusions drawn about the differences in sepsis knowledge between countries are not fully supported by a rigorous analysis due to the limitations of the sampling method and questionnaire validation. The interpretation of statistical results should be more cautious, given the potential biases introduced by the convenience .
Answer 4: We modified conclusion according to your recommendations. This study showed that nursing students in this sample still have limited knowledge of sepsis, and that there are differences between education programmes for nursing students. These findings suggest that variations in educational programs or teaching methodologies in universities included in research may contribute to the differences in students' knowledge levels.
Comment 5: Main Questions Addressed: The study adequately addresses the main question of comparing sepsis knowledge across countries. However, the methodology's limitations weaken the strength of the conclusions. Specific experiments or analyses (e.g., factor analysis for questionnaire validation, more detailed demographic analysis) should be included to strengthen the conclusions.
Answer 5: We changed in the manuscript.

Round 2
Reviewer 2 Report (Previous Reviewer 2)
Comments and Suggestions for Authors
Congratulations to the authors for their hard work and addressing most of the comments in the previous review round. However, minor issues are still present and should be properly addressed before the manuscript is good to go.
1- The figure titles should be placed below the figures, contrary to the table titles.
2- Figure legends should be clearly written, describing the figures in details.
3- My previous comment (No.3) has been partially addressed. In the figure legend of each figure, authors should describe the data based on its distribution, and the statistical method used in for each comparison.
Author Response
Dear reviewers,
we changed all according your instructions. Document attached.
Kindest regards.
Jelena i Adriano

This manuscript is a resubmission of an earlier submission. The following is a list of the peer review reports and author responses from that submission.
Round 1
Reviewer 1 Report
Comments and Suggestions for Authors
In detail, I read and analysed the manuscript Nursing Student Perspectives Related to Sepsis in Three European Countries: A Cross-Sectional Study.
The title is adequate and matches the content of the manuscript.
The topic is timely and relevant for the scientific audience because the authors analyse one of the most important global health issues – sepsis, or the knowledge of students from three different countries about sepsis. Researching the sepsis problem using this approach is significant for clinical practice and education.
However, I would have a lot of suggestions for corrections.
The abstract is not precise and does not concisely summarise the manuscript content. Correct it according to the recommendations below.
Introduction is difficult to read and follow because there is no structure. One gets the impression that the authors did a literature review without critical analysis. For example, in the introduction, the authors highlight the problem of leaving the profession and the influence of educators and managers on its prevention, and at the same time, they did not link knowledge about sepsis to the mentioned problem. Likewise, health literacy is mentioned in the introduction but is not connected to the research problem.
I recommend that the authors focus more on the problem of sepsis, what nurses’ duties are in treating sepsis, and which domain of nurses’ knowledge is insufficient based on previous studies. Then, whether and how much content is there in the nursing curriculum on sepsis. Is that enough or not? This is particularly significant because nursing education differs across the three research countries. Finally, mention recent studies on students’ knowledge about sepsis and why it is important to continue researching this problem, comparing three countries.
Methodology
Respondents and Procedure
The study design is specified. However, the method and sample are not clear.
How many faculties participated from each country? One or more?
What were the inclusion and exclusion criteria for the study?
What is the response rate in each country? For example, the number of students in Croatia and Greece significantly differs.
Has the sample size been calculated?
For the period of conducting the study, it was stated that they were during 2022-2023. Is it two years, or do you mean the school year? Was the survey conducted during the summer and winter semesters if it was a school year?
Also, how the questionnaire was distributed? On-line or paper version during class.
Have you ensured that the procedure is the same in all countries?
Research Instruments
Instrument research is not fully displayed. The authors used the questionnaire by Eitze et al.. And then they improved two items. They did not specify which two items, but refer to references 6 and 16. In citing references, only reference 6 should appear because I assume that, given that the study (ref, 6) was conducted in Croatia, the version used in this study questionnaire from that study.
According to the topic, the questionnaire was adapted for the Croatian context, but is it for Greek and Cypriot? It should be stated.
Data Analysis
Data Analysis is acceptable.
Ethical Considerations
Ethical considerations are acceptable for Croatia, but the question is whether ethical approvals have been obtained for research in Greece and Cyprus. It should be stated.
Results are presented in tabular and textual form. Correcting the title or content for tables 2 and 4 is necessary. Namely, in the text, when referring to the content of the table, it is written, „Table 2 shows the number and percentage of respondents with correct and incorrect or unsure answers to questions about sepsis. “ So, the table does not list only the percentages of correct answers.
In the discussion, the authors refer to previous research.
The study’s limitations are not identified. It should be stated and strengthened.
The conclusion is imprecisely written. Authors must state the main findings from the study objectives or research questions.
The references used are adequate.
Author Response
Thank you for your comments and opportunity to improve our manuscript Nursing Student Perspectives Related to Sepsis in Three European Countries: A Cross-Sectional Study.
Hope we addressed all your comments in the way you wanted to be. All changes in main manuscript are marked with yellow colour.
For all authors,
Adriano Friganovic and Gloria Besker

Reviewer 2 Report
Comments and Suggestions for Authors
I read with interest the study presented by Friganovic and colleagues on nursing student's knowledge about sepsis in Croatia, Cyprus and Greece. The study presents a topic of clinical significance and its presentation is of timely importance, especially in the light of the recent pandemic COVID-19.
The study is meritted with several strengths, on which the authors are commended. These include:
- The multinational approach to highlight the issue at hand.
- The use of a previously established tool, i.e., questionnaire, which would facilitate comparison with other studies using the same tool.
Nevertheless, the manuscript in its current form requires significant changes before it is ready for publication in such an internationally oriented journal. The following is recommended issues to be address:
- The title of the study states "perspectives" where it is acutally knowledge that is assessed in this study.
- The introduction needs to be rewritten with a clear focus on the importance of sepsis in clinical practice and why nursing students need to fully aware of it, rather than nursing students leaving the profession.
- The introduction does not provide proper context of the decision of choosing these countries, and what are the current difference between these countries in the nursing curriculum. The only difference mentioned, was the number of years.
- In the Methods section, please provide the full questionnaire used, either as a complete table or a supplementary. Also, please clearly explain the scoring system used in this study.
- The study tool, questionnaire, was used in three different countries. However, it is not clear in which language was this tool used, especially as these countries do not necessarily speak the same language, nor the language of instruction is similar.
- What was the rationale of including nursing from all year levels? Would not a fourth year student be more knowledgable about spesis compared to a first year student?
- Authors should point out how they determined level of statistical significance.
- It would be of great interest to demonstrate inter-level differences (year 1 vs year 2 ..etc) between countries. Such differences would significantly improve the results and highlight key lessons from which authors could develop recommendations.
- The discussion needs to significantly enhanced. For example, an entire paragraph should be included to discuss the strengths and the limitations of this study, e.g., the unequal sample size between countries. The discussion should also include how these limitations should be addressed in future research.
Comments on the Quality of English LanguageA thorough proof of the manuscript would significantly improve the readability and presentation of the manuscript.
Author Response

(The authors gave the same response as above.)

Round 2
Reviewer 1 Report
Comments and Suggestions for Authors
The authors accept all the suggestions given, so the revised version of the manuscript titled Nursing Student Knowledge Related to Sepsis in Three European Countries: A Cross-Sectional Study is now easy to read, and all parts of the manuscript are better presented, which has improved the overall quality of the manuscript.
Considering the aforementioned and the importance of the topic that the authors examined, I recommend that the manuscript should be accepted.
Author Response
Dear editor and reviewers,
Thank you for your comments and opportunity to improve our manuscript Nursing Student Perspectives Related to Sepsis in Three European Countries: A Cross-Sectional Study.
For all authors,
Adriano Friganovic and Gloria Besker

Reviewer 2 Report
Comments and Suggestions for Authors
Thank you for addressing most of the comments raised, which significantly improved the paper. However, two issues still persist and the authors' responses to these issues were unsatisfactory. These issues are:
- The introduction does not provide proper context of the decision of choosing these countries, and what are the current difference between these countries in the nursing curriculum. The only difference mentioned, was the number of years. Please clearly state key differences in terms of medium of instruction, mode of instruction, curriculum content, and the practical training.
- It would be of great interest to demonstrate inter-level differences (year 1 vs year 2 ..etc) between countries. Such differences would significantly improve the results and highlight key lessons from which authors could develop recommendations. This is a key comparison that needs to be done to stratify the differences between the comparison groups.
Comments on the Quality of English LanguageStill to be done.
Author Response
Dear editor and reviewers,
Thank you for your comments and opportunity to improve our manuscript Nursing Student Perspectives Related to Sepsis in Three European Countries: A Cross-Sectional Study.
Hope we addressed all your comments in the way you wanted to be. All changes in main manuscript are marked with yellow colour.
For all authors,
Adriano Friganovic and Gloria Besker
